# Exploring the Process of Neutrophil Transendothelial Migration Using Scanning Ion-Conductance Microscopy

**DOI:** 10.3390/cells12131806

**Published:** 2023-07-07

**Authors:** Svetlana N. Pleskova, Nikolay A. Bezrukov, Ekaterina N. Gorshkova, Sergey Z. Bobyk, Ekaterina V. Lazarenko

**Affiliations:** 1Research Laboratory of Scanning Probe Microscopy, Lobachevsky State University of Nizhny Novgorod, 603950 Nizhny Novgorod, Russia; nick_bezrukov@mail.ru (N.A.B.); e.n.gorshkova@gmail.com (E.N.G.); slbobyk@gmail.com (S.Z.B.); rinlx@mail.ru (E.V.L.); 2Department “Nanotechnology and Biotechnology”, Nizhny Novgorod State Technical University n. a. R.E. Alekseev, 603115 Nizhny Novgorod, Russia

**Keywords:** neutrophil, endothelial cells, transendothelial migration, septicopyemia, scanning ion-conductance microscopy, alteration, inflammation, *Staphylococcus aureus*, cell rigidity, NETs

## Abstract

The dynamics of neutrophil transendothelial migration was investigated in a model of experimental septicopyemia. Scanning ion-conductance microscopy allowed us to determine changes in morphometric characteristics of endothelial cells during this process. In the presence of a pyogenic lesion simulated by *Staphylococcus aureus*, such migration was accompanied by both compensatory reactions and alteration of both neutrophils and endothelial cells. Neutrophils demonstrated crawling along the contact sites between endothelial cells, swarming phenomenon, as well as anergy and formation of neutrophil extracellular traps (NETs) as a normergic state. Neutrophil swarming was accompanied by an increase in the intercellular spaces between endothelial cells. Endothelial cells decreased the area of adhesion to the substrate, which was determined by a decrease in the cell projection area, and the cell membrane was smoothed. However, endothelial cell rigidity was paradoxically unchanged compared to the control. Over time, neutrophil migration led to a more significant alteration of endothelial cells: first, shallow perforations in the membrane were formed, which were repaired rather quickly, then stress fibrils were formed, and finally, endothelial cells died and multiple perforations were formed on their membrane.

## 1. Introduction

Transendothelial migration of neutrophils is an important step in the development of the body’s general defensive reaction of inflammation. The interaction between neutrophils and endothelial cells occurs through different adhesion modules, and it is a fundamental event determining whether neutrophil transmigration occurs either paracellularly or transcellularly. The process of extravasation manifests the plasticity of neutrophil functions [1]. Biochemical aspects of the transendothelial migration process are well studied [2], but the introduction of new research methods significantly enriches the knowledge of the process features and possibilities of its regulation. In particular, using intravital imaging, a group of researchers [3] discovered an unexpected role for platelets in neutrophil crawling. It was shown that recruited neutrophils scan activated platelets, integrate endothelial and platelet signals before inflammation begins, and interaction with activated platelets leads to neutrophil polarization and redistribution of receptors on them. A group [4] using a similar technique discovered an important phenomenon of micro- and macroleakage of liposomes accompanying neutrophil extravasation. In particular, it was shown that transendothelial migration of neutrophils causes faster accumulation of liposomes in tumor tissues.

Despite significant progress in understanding the molecular, biochemical and morphological aspects of transendothelial migration, there are significant gaps in understanding the mechanics of the process, in particular, changes in endothelial cell and neutrophil rigidity during migration. Currently, fundamentally new methods of investigation have emerged that allow to simultaneously (1) carry out high-resolution observations of the migration process without using fixation, under conditions as close to native as possible; (2) simultaneously study changes in the stiffness of cells during observation; (3) in cases of integration with confocal or multiphoton microscopy, observe redistribution of cell receptors during diapedesis. Such methods include scanning ion-conductance microscopy (SICM), which gives a resolution comparable to scanning electron microscopy, but allows observations in the native environment, i.e., the cellular process rather than a fixed sample is studied in real time [5]. Currently, there are two main problems complicating high-resolution imaging of neutrophil transendothelial migration: (1) the necessity of fixation of endothelial cells on the matrix, simultaneously ensuring their proliferation and normal viability, as well as the capability of neutrophil granulocyte migration; and (2) development of a favorable environment, supporting the viability of both neutrophils and endothelial cells simultaneously during the experiment [6].

Attempts were made to study neutrophil migration using high-resolution atomic force microscopy (AFM). This continues to be used as an auxiliary tool in the study of transendothelial migration, as a rule, to assess endothelial cell rigidity or determine adhesion strength [7,8]. It was also used for the same purpose by Martinelli et al. in 2014, who achieved interesting results in imaging transendothelial migration. Combining fluorescence, electron and atomic force microscopy, the researchers visualized “bulges” mediating the process of diapedesis through “soft” regions providing the least resistance to leukocyte passage through the endothelium. The developed system of intravital microscopy visualized human and rat T-cells actively forming multiple invadopodia-like bulges in the area above the endothelial cell nucleus at the moment preceding the formation of the transcellular pore. The process of horizontal migration of leukocytes along the endothelial cells was also visualized, indicating selection of the area for migration. After a 10 min residence on the human pulmonary endothelium or rat brain endothelium (MVEC), the cells were fixed for further analysis with electron microscopy [9]. The only significant limitation of imaging in this system was the matrix (adhesive Petri dishes), which prevented full migration. Additionally, the method requires a large number of contrast agents and provides limited resolution. Thus, a system that would accurately simulate three-dimensional structure (blood-endothelial cell-tissue) and simultaneously provide the possibility of high-resolution study and determination of cell rigidity does not currently exist.

We have designed and tested a two-chamber system, allowing long-term observation of endothelial cells surrounded on all sides by nutrient medium. It was shown that the AFM method does not allow observations in such a system due to a pronounced mechanical impact on the membrane on which the endothelial cells were grown, while the SICM method provides an opportunity to conduct dynamic studies up to the onset of endothelial cell apoptosis. Optimal conditions for long-term observation were selected in [10].

The aim of this work was to study the process of transendothelial migration in the developed two-chamber system by high-resolution SICM and evaluate the morphological changes of cells during this process.

## 2. Materials and Methods

### 2.1. Culture of EA.hy926 Cells

EA.hy926 cell line was cultured in Dulbecco’s modified eagle medium: nutrient mixture F12 (DMEM-F12) containing 10% inactivated fetal calf serum, 100 µg/mL streptomycin, 100 units/mL penicillin, 8 mM L-glutamine, HAT (50 µM hypoxanthine, 0.2 µM aminopterin, 8 µM thymidine) (all from PanEco, Moscow, Russia). Cells were cultured in a CO_2_ incubator Portable Mini NB 203M (N-Biotek, Bucheon-si, South Korea) at 37 °C, 5% CO_2_. Monolayer disaggregation was performed using 0.25% trypsin-EDTA solution. Cells were passaged every 3–4 days when the cultures reached 90–100% confluence according to the conventional technique [11], causing monolayer disintegration by 3 min exposure in trypsin-EDTA solution (PanEco, Russia). The cells at passages 3–25 at a concentration of 2.5 × 10^5^ cells/mL were used in the experiments.

### 2.2. Isolation of Neutrophil Granulocytes

The neutrophils from healthy donors were isolated from heparinized (25 U/mL) venous blood on a double Ficoll-Trazograph (Dia-m, Moscow, Russia; JBCPL, Mumbai, India) gradient (ρ = 1.077, ρ = 1.110, 400 g, 40 min) [12], then washed twice with sterile buffered saline (400 g, 3 min). Neutrophils were used at a final concentration of 2 × 10^5^ cells/mL in the Hanks’ balanced salt solution (HBSS) contained 0.01 M 4-(2-hydroxyethyl)-1-piperazineethanesulfonic acid (HEPES) with pH 7.3. The study was approved by the Commission on Bioethics of the N.I. Lobachevsky State University of Nizhny Novgorod (created on 11.11.16, order for the creation No. 497-OD), protocol No. 9, dated 17.07.17.

### 2.3. S. aureus Culturing

Cultivation of *S. aureus* 2879 M strain is described in [13]. Briefly, *S. aureus* 2879 M was cultured (37 °C, 24 h) in meat-peptone “GRM” agar (Obolensk, Russia); cells were washed from the medium then centrifugated 3 times with sterile PBS (1800 g, 10 min), resuspended in PBS, and adjusted to the optical density of 0.75 (λ = 670 nm), which corresponded to 1 × 10^9^ cells/mL, using spectrophotometer SPECS SSP 705 (Spectrophotometric Systems, Moscow, Russia). Bacteria were used in a final concentration of 2.5 × 10^7^ cells/mL in the same medium as neutrophils.

### 2.4. Cultivation of a Monolayer of Endothelial Cells on the Cell Culture Inserts

For the formation of the monolayer cells were plated in DMEM-F12 medium at a concentration of 2.5 × 10^5^ cells/mL on 6.5 mm in diameter cell culture polycarbonate inserts with adhesive coating and incubated in 35 mm Petri dishes (Corning, NY, USA) for 72 h (37 °C, 5% CO_2_). Monolayer formation was monitored using an Axio Vert.A1 (Carl Zeiss, Oberkochen, Germany) optical microscope.

### 2.5. Study of Transendothelial Migration of Neutrophil Granulocytes in the Model of Experimental Septicopyemia

To simulate septicopyemia (purulent foci in tissues), 3D-printed inserts fixing membranes in a polystyrene Petri dish were created [10]. A model of septicopyemia was used as follows: *S. aureus* 2879 M bacteria were added to the lower chamber in HBSS medium + 0.01 M HEPES (pH 7.3), then the insert with the EA.hy926 monolayer was washed with the same medium and gently placed on the interface between the lower and upper chambers. HBSS + HEPES was also added to the upper chamber to properly coat the cells. The choice of the medium for the study was due to the fact that it provided the maximum viability of both cell types (both neutrophils and endothelial cells) [10]. A series of control scans to check the morphology and rigidity of endothelial cells were taken and, after that, neutrophil granulocytes were introduced into the upper chamber and their migration through the endothelial cell monolayer was studied using SICM (Figure 1).

### 2.6. Scanning and Measuring of Cell Rigidity by the Method of Scanning Ion-Conductance Microscopy

The cells were scanned using a scanning ion-conductance microscope (ICAPPIC Ltd., London, UK). Scanning was performed in a hopping mode with a fall rate of 100 nm/ms at a bias potential of 200 mV using nanopipettes as probes. Nanopipettes with a characteristic inner tip radius of about 25–50 nm were obtained using a laser puller P-2000 (Sutter Instruments, Novato, CA, USA). The ion current was measured using a MultiClamp 700 B amplifier (Molecular Devices, Wokingham, UK). The stress that the nanopipette induces on a cell was calculated from the gap size in terms of the ion current decrease at two different set points 0.5% and 2%. The resulting images and calculations were processed using SICMImageViewer software (ICAPPIC Ltd., UK).

### 2.7. Statistical Analysis

The statistical analysis was performed using Origin 8.0 software (Origin Lab Corparation, Northampton, MA, USA). Boundaries of normal distribution were determined for quantitative indicators of samples using Shapiro–Wilk test. Since the distributions met the measure of normality, we determined the mean and standard deviation. Statistical significance of differences between the two samples was determined with Student’s *t*-test.

## 3. Results

### 3.1. The Endothelial Cell Monolayer for Investigation of Neutrophil Transendothelial Migration

Since it is known that capillary walls, through which transendothelial migration is mainly realized, consist of only a few thin layers of squamous endothelial cells [14,15], at the first stage it was important to obtain a monolayer of EA.hy926 culture. The optimal conditions for monolayer formation without culture overgrowth were an initial culture cell concentration of 2.5 × 10^5^ cells/mL and an incubation time of 48 to 72 h. The monolayer of endothelial cells grown on the membrane that separates the lower chamber with bacteria and the upper chamber with neutrophils is shown in Figure 2.

The different ages of the culture are well visualized in Figure 2A: older cells are flatter and better adherent, younger cells are elevated and more globular in shape.

### 3.2. Morphometric Studies of EA.hy926 Culture Monolayer by Scanning Ion-Conductance Microscopy

Since there was a pronounced dispersion not only in the age, but also in the size of EA.hy926 cells, morphometric measurements of endothelial cells were carried out. The cell projection area and cell volume were measured. The results of the measurements are summarized in Table 1. The mode of measuring the distribution of ionic currents allows us to simultaneously obtain topographic scans with the possibility of morphometric analysis and build a map of rigidity distribution, i.e., the distribution of cell membrane stiffness. Therefore, Table 1 also presents the results of measuring average values of endothelial cell rigidity.

### 3.3. Changes in Neutrophil Granulocytes during Transendothelial Migration

In the model of experimental septicopyemia (Figure 1), there was a bacterial suspension in the lower part of the analytical chamber which released chemoattractants for neutrophils introduced into the upper chamber. An overwhelming number of neutrophils, after being introduced into the upper chamber, first settled on the monolayer of EA.hy926 culture and then migrated through the endothelium into the lower chamber. The formation of aggregates of several neutrophils, rather than the migration of single cells, was most frequently detected. Single neutrophils (Figure 3A) forming the pore facilitated the passage of large neutrophil aggregates through this pore (Figure 3B).

The presence of a high concentration of bacteria in the lower chamber likely provoked a state of anergy in some neutrophils, in which they remained immobile, attached to the endothelial cells and did not change morphology during 90 min of observation (Figure 3C).

On the contrary, some neutrophils actively crawled along the surface of the endothelial cells (Figure 4). Initially, the establishment of intercellular contacts between neutrophils and endothelial cells was observed (about 7 min), after which the neutrophils apparently primed and began to actively crawl. Neutrophil crawling was noted exclusively in the intercellular contacts between neighboring endothelial cells, although the full-time observation of a neutrophil (30 min) showed that the cell could change its speed of movement. The average speed of neutrophil migration along endothelial cell intercellular contacts averaged 2 μm/min.

For one neutrophil, a very interesting phenomenon was observed: the cell formed a neutrophil extracellular trap (NET)-like structure, which was connected to the endothelial cell fibril. For some time, the neutrophil oscillated in this bound state, and then cell death by NETosis occurred, and the NET leftovers remained on the endothelial cell surface (Figure 5).

### 3.4. Endothelial Cell Changes during Transendothelial Migration of Neutrophils

The passage of neutrophils through the endothelial cell monolayer was accompanied by significant changes in the endothelial cells. The main changes in morphology were associated with the transition of endothelial cells from a flattened state into a more rounded one, with partial loss of cell contact area with the polycarbonate substrate (Figure 6), which was also reflected in a statistically significant decrease in cell projection area, although cell volume did not change (Table 1). Endothelial cell rigidity also did not undergo statistically significant changes during neutrophil migration. The surface of the cells was smoothed.

Long-term observation of endothelial cells during and after neutrophil migration revealed signs of endothelial cell alteration. In the process of neutrophil transmigration, a shallow perforation could form in the endothelial cell membrane (Figure 6A). However, at the initial stages of migration, endothelial cell regenerative capabilities were sufficient for complete repair of the resulting damage and, with time, the hole was completely closed and the membrane regenerated. Increasing the observation time up to 120 min showed an increase in the alteration damage on the endothelial cell side, expressed in the formation of stress fibrils by the cells (Figure 6B). In some experiments, the alteration of endothelial cell membranes went very deep and numerous perforations were formed on the surface of the cells, which did not regenerate (Figure 6C).

## 4. Discussion

One of the important factors in the effective response of neutrophils to the formation of purulent foci in severe sepsis is rapid diapedesis into the affected tissues. However, in mice experiments, it has been shown that lethal intraperitoneal inaculum of *S. aureus* causes a 92% decrease in neutrophil migration [16]. In our study, in the model of experimental septicopyemia, the use of high-resolution scanning ion-conductance microscopy revealed the alteration of both neutrophil granulocytes and endothelial cells during transendothelial migration.

In particular, it was shown that the presence of *S. aureus* in the lower compartment of the two-section analytical chamber, simultaneously used as a source of chemoattractants and as a model of the pyemic focus, can provoke a neutrophil anergy state, when the cells remain on the endothelial cell surface for a long time without migrating and without changing their morphology (Figure 3C). The formation of large intercellular spaces between the endothelial cells and neutrophils migration in large aggregates in this case can be considered both as alteration and as a compensatory mechanism ensuring the transmigration of more neutrophils (Figure 3A,B). Most likely, the phenomenon of micro- and macroleakage of vessels after neutrophils escape into the extravasal space, discovered by the authors of [4], is caused precisely by the increase in the intercellular spaces between endothelial cells under the influence of swarming neutrophils. Swarming (aggregating) neutrophils have been shown to self-reinforce the chemotactic attraction of additional neutrophils to the sites of bacterial invasion, due to the release of chemoattractants acting on other cells [17], and the increase of intercellular spaces with fluid exudation is a long and well-known sign of inflammation, contributing to additional migration of humoral and cellular factors to the site of infection [18]. However, high concentrations of chemoattractants can attenuate cellular responses through a process called GPCR desensitization, and neutrophil swarming can lead to impaired bacterial clearance [17]. Therefore, the formation of neutrophil aggregates and their swarming in this case can also be considered as an alteration.

The most dangerous process is the formation of NETs by neutrophils (Figure 5), as it plays a significant role in the pathogenesis of sepsis [19,20] and can contribute to the development of a severe complication of sepsis—DIC syndrome [21]. Normally, NET formation within vessels and initial adhesion of neutrophils on endothelial cells is not desirable [22]. At the same time, it has been shown that activated endothelial cells can provoke the formation of NETs [23]. In our experiments, endothelial cells were activated by *S. aureus* located in the lower chamber. In turn, NETs significantly activate human endothelial cells by increasing the expression of both mRNA and VCAM-1 and ICAM-1 [24].

In our experiments, endothelial cell activation was accompanied by alteration processes. On the one hand, the increase in intercellular spaces, which we detected as a statistically significant decrease in the cell projection area, can be considered as an adaptive mechanism promoting neutrophil transmigration. This version is also supported by the absence of differences in the rigidity of the membrane–cytoskeleton complex of endothelial cells before and after migration. On the other hand, during the long-term observation there were also clearly expressed signs of alteration: formation of shallow perforations in the membrane, its flattening, formation of stress fibrils and multiple perforations during endothelial cell death. In the in vivo system, regenerative mechanisms are actively involved in maintaining the homeostatic integrity of the vascular wall. This may occur as a result of changes in the transcriptional profile of endothelial cells and their acquisition of a significant proliferative potential [25]. Damage repair may require elongation and migration of resident endothelial cells [26]. Usually, in case of vascular endothelium damage, the source of regeneration is the resident endothelial cells of precisely those organs (liver, lungs) where the damage occurred [27,28]. On the part of endothelial cells, as well as on the part of neutrophils, the combination of reparative and alteration processes and the predominance of one or another of them depends, first of all, on the severity of the septic process.

Thus, real-time observation of the transendothelial migration process in the model of experimental septicopyemia reveals a complex mixture of alterative and compensatory mechanisms ensuring effective diapedesis of neutrophils into the extravasal space, where the pyogenic lesion is formed.

## Figures and Tables

**Figure 1 cells-12-01806-f001:**
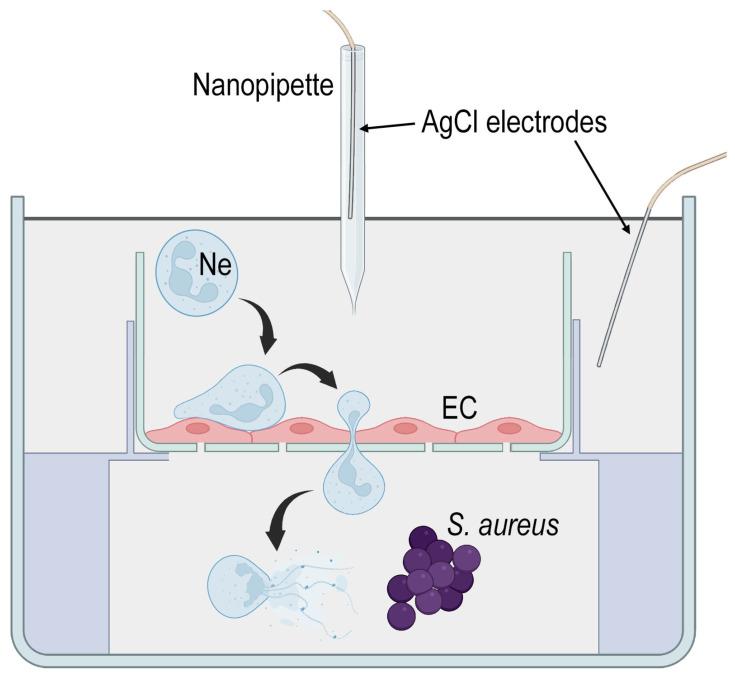
The schematic arrangement of the two-section analytical chamber: in the lower compartment, bacteria are absorbed by neutrophils; the compartments are separated by a polycarbonate membrane on which a monolayer of endothelial cells is grown, neutrophil granulocytes are introduced into the upper chamber, and then they migrate through a layer of endothelial cells following the chemoattractant gradient.

**Figure 2 cells-12-01806-f002:**
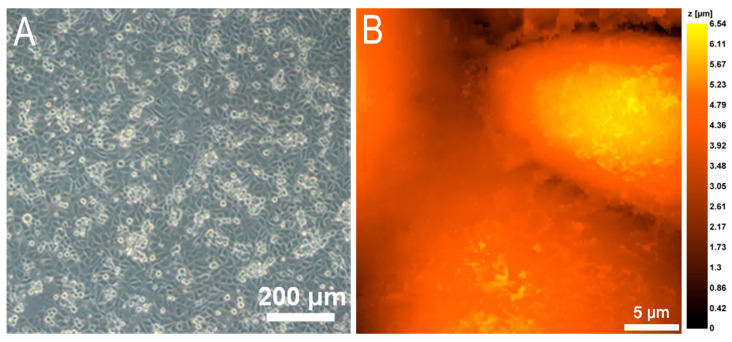
A monolayer of EA.hy926 cells on a polycarbonate membrane with a pore diameter of 3 μm. (**A**) Light microscopy (10×, Olympus, Tokyo, Japan). (**B**) Scanning ion-conductance microscopy (ICAPPIC Ltd., UK).

**Figure 3 cells-12-01806-f003:**
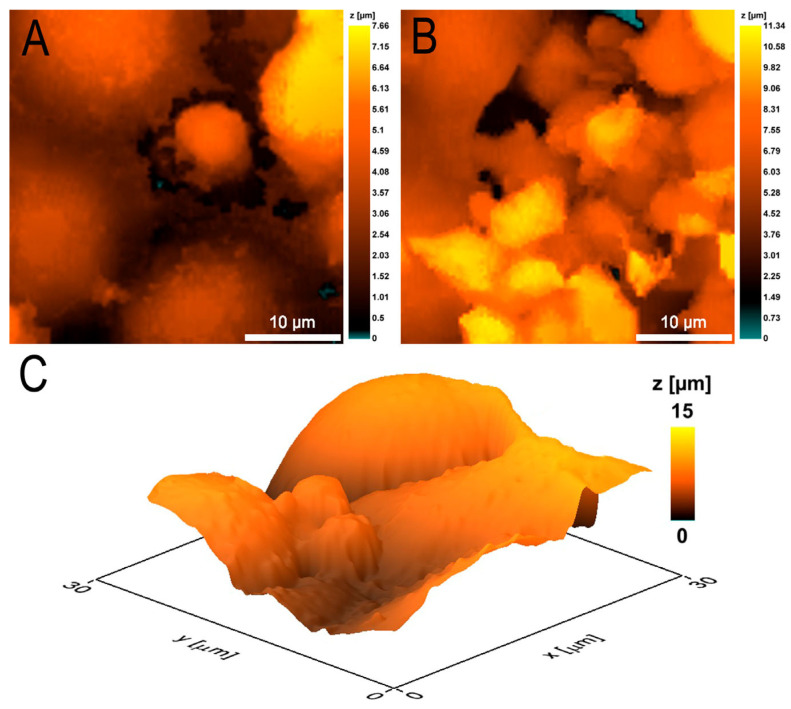
Different behavior of neutrophils during transendothelial migration. (**A**) Formation of a pore between endothelial cells by a single neutrophil. (**B**) Passage of neutrophil aggregates through the formed pore. (**C**) Neutrophils go into anergy state, do not migrate through interendothelial space, and do not change morphology for a long time.

**Figure 4 cells-12-01806-f004:**
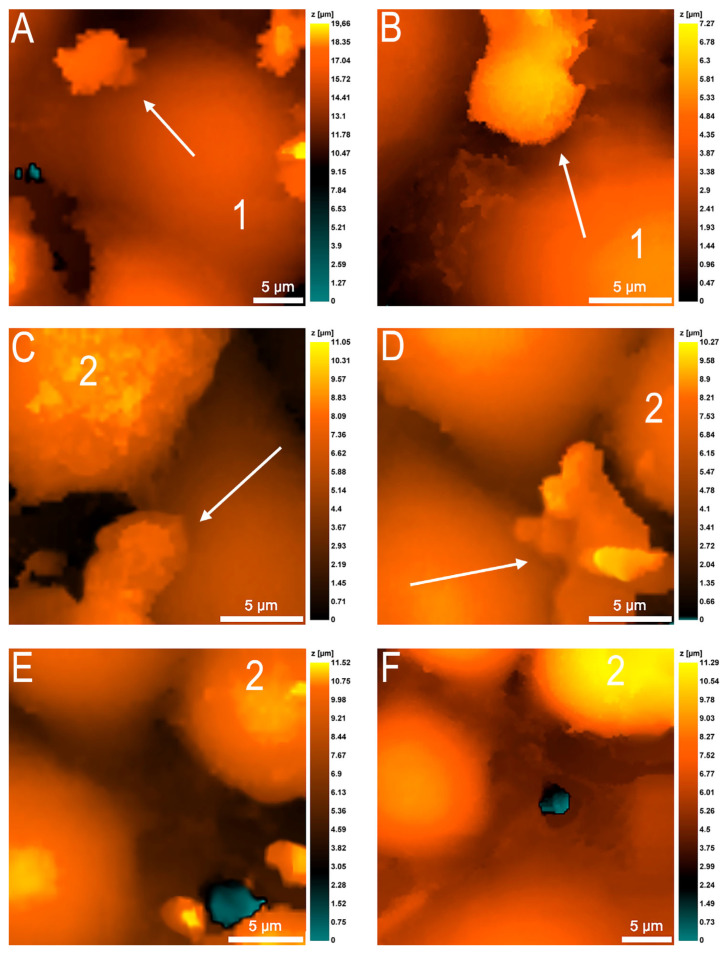
Active neutrophil migration (marked by arrows) along the surface of the endothelial monolayer: the neutrophil moved exclusively in the areas of interendothelial contacts; sometimes neutrophil pseudopodia interacted with the surface of endothelial cells (**A**–**F**). Endothelial cells are marked with numbers to facilitate navigation.

**Figure 5 cells-12-01806-f005:**
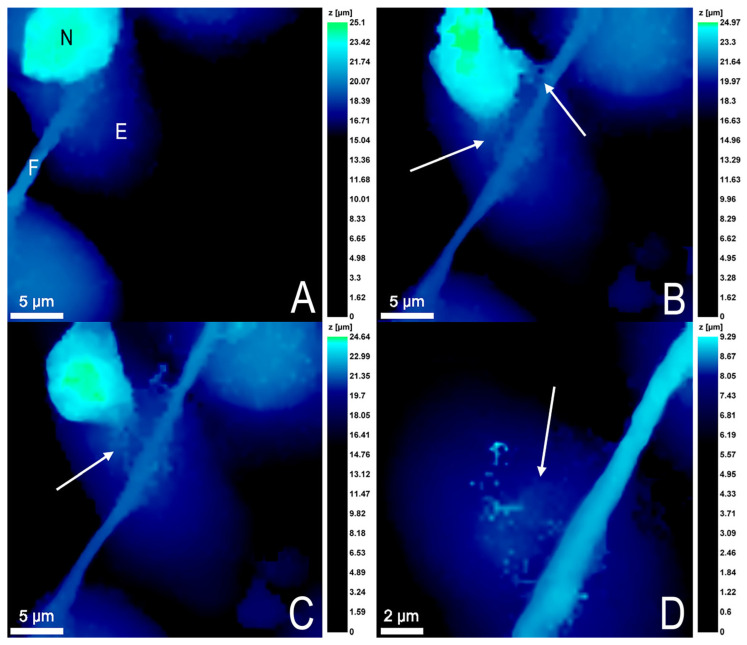
Formation of a neutrophil extracellular trap (NET)-like structure by neutrophil. (**A**) The neutrophil (N) binds to the endothelial cell (E) fibril (F). (**B**) The neutrophil forms a NET and remains bound to the endothelial cell fibril for some time and oscillates on it. (**C**) The neutrophil gradually decreases in volume, then dies by NETosis. (**D**) Remains of a NET structure are formed on the endothelial cell surface. The arrows indicate NET-like structures.

**Figure 6 cells-12-01806-f006:**
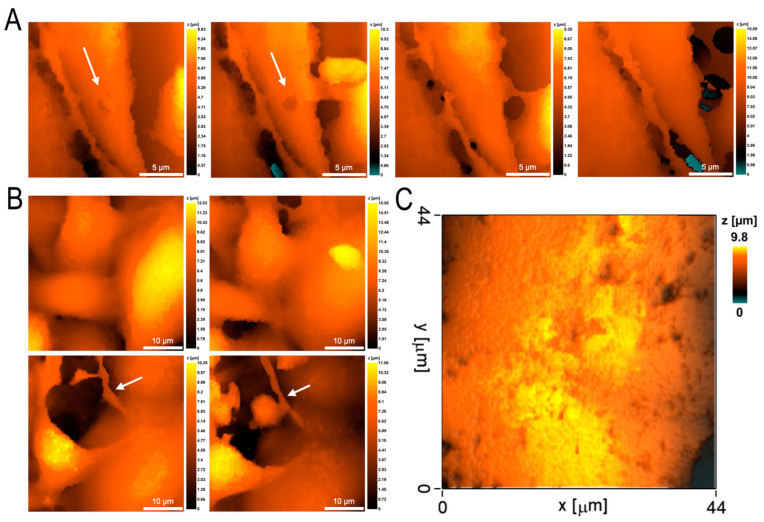
Endothelial cell alteration during and after neutrophil migration in the model of the experimental septicopyemia. (**A**) Formation of shallow perforation (marked by arrows) with its subsequent regeneration. (**B**) Formation of stress fibrils by endothelial cells (marked by arrows) 120 min after the beginning of observation; neutrophil interaction with such a stress fibril. (**C**) Endothelial cell death with formation of multiple membrane perforations.

**Table 1 cells-12-01806-t001:** Morphometric characteristics and endothelial cell stiffness before and after migration of neutrophil granulocytes into the lower chamber (*n* = 20).

Characteristics	Before Migration	After Migration
Cell projection area, μm^2^	422.3 ± 116.7	354.6 ± 105.2 ^1^
Cell volume, μm^3^	1620.8 ± 580.2	1697.2 ± 663.3
Young’s modulus, Pa	758.5 ± 158.1	711.9 ± 129.3

^1^ Statistically different from control, *p* < 0.001.

## Data Availability

Data are available from the corresponding author.

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
