# Peer review of "Exploring the Process of Neutrophil Transendothelial Migration Using Scanning Ion-Conductance Microscopy"

_cells, 2023, doi:10.3390/cells12131806_

Round 1

Reviewer 1 Report

See attached. 

The manuscript is well written and clear. 

Author Response

Dear colleague, thank you for your work with our article. The responses to comments are provided via attached PDF

Reviewer 2 Report

- Please use the full names of abbreviations before using abbreviations.

- There are some typo errors. e.g., CO2 not CO2.

- Although hour, hr, and h are correct, the authors should be used one of them.

- Could the authors explain what is the difference between the current study and reference 11 "Pleskova et al., 2023".

- Some references are old references and there are new versions of the manuscript. Please use the new papers instead of the old papers.

There are some typo errors and the work needs moderate English editing.

Author Response

(The authors gave the same response as above.)

Reviewer 3 Report

The research report by Pleskova SN and colleagues entitled “Exploring the process of neutrophil transendothelial migration using scanning ion-conductance microscopy” submitted for publication to “Cells” presents morphometric data (SICM) on the interaction of endothelial cells and neutrophil upon challenge with Staphylococcus aureus (S.a.) in “real-time”. They employed and optimized scanning ion-conductance microscopy, SICM, and thus acquired dynamic topographic information on the endothelial cell´s membrane rigidity. In general, the author´s provide a good rationale for their experimental approach, methods are sufficiently described (except for S.a. numbers, alive?) and the authors interpretation proves attention to details. It is clear that SICM is not a high-throughput methodology and thus this research report can be considered as “case study”. Together, I support a publication in this form.

Minor comments and suggestions:
1) As the authors mention the role of platelets in the context endothelial cells (EC´s) - neutrophils, it would be interesting to see how EC´s would behave in presence of platelets, and in the combination platelets +neutrophils. In addition, modulation of EC´s membrane in response to the pathogen S.a. is not described in this study. Thus, it might be worth to also diversify i.e. towards different S.a. strains and even pathogens.

2) Along this line, having SICM established, it is still advisable to provide and confirm some phenomena via ultrastructure-electron microscopy.   

3) the term “endotheliocytes” is rarely used in the basic research literature, the author´s may consider to re-phrase to “endothelial cells” throughout the manuscript.

Author Response

(The authors gave the same response as above.)
